# Evaluation of Systemic Antifungal Prescribing Knowledge and Practice in the Critical Care Setting among ICU Physicians and Clinical Pharmacists: A Cross-Sectional Study

**DOI:** 10.3390/antibiotics12020238

**Published:** 2023-01-23

**Authors:** Sahar Mohamed Ibrahim, Nosiyba Adlan, Sufyan Mohammed Alomair, Ibrahim Butaiban, Ahmed Alsalman, Abdulmajeed Bawazeer, Monahi Alqahtani, Dalia Mohamed, Promise Madu Emeka

**Affiliations:** 1Department of Pharmacy Practice, College of Clinical Pharmacy, King Faisal University, Al-Ahsaa 31982, Saudi Arabia; 2Department of Pharmacy, Almoosa Specialist Hospital, Al-Ahsaa 31982, Saudi Arabia; 3Department of Pharmaceutical Sciences, College of Clinical Pharmacy, King Faisal University, Al-Ahsaa 31982, Saudi Arabia

**Keywords:** antifungal, knowledge, prescribing, ICU, HCPs, *Candida*

## Abstract

Management of invasive fungal infections (IFI) and subsequent treatment choices remain challenging for physicians in the ICU. Documented evidence shows increased practice of the inappropriate use of antifungal agents in the ICU. Continuous education of healthcare providers (HCPs) represents the cornerstone requirement for starting an antifungal stewardship program (AFS). This study aimed at evaluating knowledge gaps in systemic antifungal prescribing among physicians and clinical pharmacists in a critical care setting. A cross-sectional, multi-center, survey-based study was conducted in five tertiary hospitals located in Al-Ahsaa, Saudi Arabia between January and May 2021. A self-administered questionnaire was distributed among the targeted clinicians. A total of 63 clinicians were involved (65.5% ICU physicians and 34.5% clinical pharmacists). It was noted that a minority of the participating HCPs (3.2%) had overall good knowledge about antifungal prescribing, but the majority had either moderate (46%) or poor (50.8%) knowledge. The difference in overall knowledge scores between the ICU physicians and the clinical pharmacists (*p* = 0.925) was not significant. However, pharmacists showed better scores for the pharmacokinetics of antifungal therapy (*p* = 0.05). This study has revealed a significant gap in the knowledge and practice of clinicians as regards prescribing antifungal therapy in our area. Although the results cannot be generalized, the outcome of this study has exposed the need for a tailored training program essential for carrying out an AFS program.

## 1. Introduction

Fungal infections account for about 20% of documented intensive care unit (ICU) infections, of which *Candida* infections represent approximately 70 to 90% of total invasive fungal infections (IFIs) [1]. The incidence of candidemia has been rising over the decades due to an increase in the number of immunocompromised and critically ill patients [2]. Along with the steady growth of fungal infections among critically ill patients, there has been a relative rise in non-*albicans Candida* infections, these species exhibit resistance to azole antifungals and represent more challenging clinical situations [3,4]. In Saudi Arabia, studies have reported a high frequency of non-*albicans* candidaemia and an alarming rise of fluconazole-resistant *Candida parapsilosis* [4,5]. The antifungal resistance coinciding with a change in the epidemiologic pattern of candidemia identified in one study was described as alarming [5]. An outbreak report described the emergence of a cluster of *Candida auris* cases in the ICU of King Khalid Hospital [6].

IFIs are associated with a substantial risk of mortality and morbidity among the critical care population. Invasive candidiasis (IC), which is a major contributor to IFIs, is associated with a high mortality rate reaching 40–60% [7]. These types of infections impose a substantial financial burden on the health system owing to an increased ICU length of stay, the need for expensive antifungal medications, and the overall increased consumption of hospital resources [8,9].

Management of IFIs and subsequent treatment choices remain challenging for physicians in the ICU [7,8]. Studies have shown that early initiation of empiric antifungal treatment improves the prognosis of IC [8,10]. On the other hand, many uncertainties surround the diagnosis of IC, which can potentially delay timely antifungal treatment [8]. Although blood culture remains the gold standard for the diagnosis of IC, its sensitivity is variable (21–71%) [11]. Unlike susceptibility testing for bacterial pathogens, some institutions lack antifungal susceptibility testing, which further complicates the appropriate antifungal choice. These difficulties facing accurate diagnosis have led to increased use of empiric antifungal therapy, especially in critically ill patients. A cross-sectional cohort study conducted in France and Belgium showed that systemic antifungal therapy was administered to 7% of all patients admitted to an intensive care unit (ICU), with only one-third of them having a proven IFI [12].

Documented evidence shows an increased practice of inappropriate use of antifungal agents in the ICU [13,14,15,16,17]. In a study conducted in a tertiary hospital in Madrid, Spain, antifungals were unnecessary in 16% of cases, most of which were consumed in the ICU. The overall prevalence of inappropriate antifungal use was 57%. Other aspects of antifungal prescription like drug selection, drug dosing, targeted therapy, and length of therapy were also inadequate [13]. A retrospective single-center cohort study concluded that the inappropriate antifungal therapy of *Candida* bloodstream infections was common and resulted in adverse clinical outcomes, substantial prolongation of hospital length of stay (LOS), and an increase in hospital costs [14]. Another retrospective chart review conducted in a teaching hospital in Saudi Arabia found that two-thirds of the caspofungin definitive therapy prescriptions were not appropriate and incorrect dosing led to a higher mortality rate [15]. Hence, there is a growing need to implement an antifungal stewardship program (AFS) [13,18,19]. Continuous education of healthcare providers, namely, ICU physicians and clinical pharmacists about IFI, their health burden, appropriate timely diagnosis, and management represents the cornerstone requirement for starting an AFS program [18].

A European survey-based study conducted in four countries to assess the knowledge and practice of European prescribing physicians in important aspects of diagnosis, prophylaxis, and antifungal treatment of IFIs, identified a serious lack of knowledge in this area [20]. Another cross-sectional multicenter survey-based study conducted across seven tertiary hospitals in Nigeria to assess the knowledge, awareness, and practice of Nigerian resident doctors regarding the diagnosis and management of invasive fungal infections, confirmed the existence of knowledge gaps [21]. These knowledge gaps among Nigerian residents were profound and were likely to impact negatively on patients with IFI as the study described [21]. Both studies recommended targeted educational programs and the Nigerian study further advised on a revision of the postgraduate medical education curriculum [20,21].

In Saudi Arabia, some studies were published to assess practitioners’ knowledge and approach toward antibiotic prescribing or antimicrobial stewardship [22,23,24]; however, studies focusing on evaluating knowledge gaps in systemic antifungal prescribing in a critical care setting are scarce. This knowledge gap in Saudi Arabia offers a unique and attractive research area to be explored. We believe this is the first study in Saudi Arabia and the Middle Eastern region to investigate clinicians’ prescribing knowledge and practice regarding antifungal therapy in a critical care setting.

The main objective of this study was to evaluate ICU physicians’ and clinical pharmacists’ knowledge and approach to prescribing antifungal agents, starting from the suspicion of IFIs and going through prophylactic, empiric, and definitive use. We also evaluated their choice of appropriate agents for treatment and to what extent this choice complies with the latest published guidelines. We hope that exploring these knowledge gaps may help identify potential flaws in current practice and assist in developing training programs for systemic antifungal prescribers.

## 2. Results

### 2.1. Socio-Demographic Characteristics of Participants

This study involved 63 healthcare providers (HCPs) (42 ICU physicians and 21 clinical pharmacists). As described in Table 1, the majority were working in the private sector (54%), and 28.6% in the Ministry of Health (MOH) hospitals. Regarding their job titles, about 40% were ICU specialists, 22% were ICU clinical pharmacists, 14% were ICU residents, and ICU consultants constituted about 10% of the participating clinicians. In addition, 50.8% had five to fifteen years of work experience, about 20% had less than three years, and only 14% had more than fifteen years of work experience.

### 2.2. Participants’ Responses to Questions about Empiric Antifungal Therapy

Table 2, Section A, details the assessment of the knowledge and practices of the clinicians about antifungal therapy. The section included 11 questions, the first five of which assessed clinicians’ knowledge and practices about empiric antifungal therapy in the ICU. According to the results of question one, which provided four case scenarios to assess the decision on starting empiric antifungal therapy, the majority (74.6%) of the participants adequately chose to immediately start empiric antifungal therapy for ICU patients who were still febrile after administration of broad-spectrum antibiotics; however, only 23.8% would start empiric antifungal therapy for dialysis patients exhibiting signs of septic shock (Figure 1). These two clinical situations necessitated the initiation of antifungal therapy. On the other hand, 33.3% and 22.2% of participants’ responses chose to start upon a positive β-D-glucan test and heavy growth of *Candida* species in urine culture, respectively, even though these treatment choices represent inappropriate utilization of antifungal therapy (Figure 1).

Pertaining to question 2, as shown in Table 2, those who spared the review of prior azole exposure for stable patients only constituted a minority (6.3%), but the majority (60%) mistakenly thought it should be checked for most patients. About one-quarter (25%) of clinicians stated that they face difficulties reviewing patients’ histories to check for prior azole exposure. Question 3 assessed the first-line choice of empiric antifungal agent in case of suspected invasive candidiasis in a critically ill non-neutropenic patient. More than two-thirds of clinicians (68.3%) correctly identified echinocandin as the first-line choice (Figure 2). Question 4 assessed the appropriate duration of antifungal therapy and 44.8% of clinicians correctly agreed that it requires 2 weeks of antifungal treatment even if the patients showed improvement (Figure 2). Question 5 investigated clinicians’ actions in case of a lack of response to antifungal therapy. Appropriately, more than three-quarters of clinicians (74.6%) agreed that antifungal therapy should be stopped if the patient showed no clinical response to the empiric antifungal treatment for 4 to 5 days (Figure 2).

### 2.3. Participants’ Responses to Questions about Targeted and Prophylactic Antifungal Therapy

As shown in Table 2, Section A, questions 6–10 assessed HCPs’ knowledge and practice about definitive or targeted antifungal therapy. Question 6 assessed the practice of de-escalating empiric echinocandin to fluconazole upon blood culture results revealing *Candida albicans*. Based on the HCPs’ responses, we found that about 57% were sure that the antifungal therapy should be de-escalated to fluconazole if the patient improved after administration of an echinocandin such as caspofungin, with blood culture results revealing *Candida albicans* sensitive species (Figure 3); whereas, about 29% would still proceed unnecessarily with echinocandins. In a similar situation, question 7 assessed HCPs’ practice in case of patient improvement on empiric echinocandin and blood culture growing *Candida glabrata* species, where echinocandins should be pursued. The responses revealed that 54% believe that they should continue with an echinocandin (Figure 3). Nevertheless, about 27% were not sure as they have not come across *Candida glabrata-*induced infections. As for *Candida krusei* isolated from blood cultures in question 8, 54% would still favor continuing with echinocandin as long as the patient improved; whereas, about 27% were not sure about the appropriate action. Only 3.2% would de-escalate to voriconazole even though this is the appropriate choice (Figure 3). Likewise, in question 9, which assessed clinicians’ actions in case of confirmed candidemia, 57% were sure that it requires immediate antifungal treatment for any ICU patient with confirmed candidemia (Table 2). Logically, neither waiting for the identification of *Candida* species nor waiting for a repeat culture are accepted practices. Question 10 assessed targeted therapy for invasive aspergillosis, where 38% of clinicians agreed that voriconazole should be used if any case of invasive aspergillosis comes across in their practice (Table 2). Pertaining to the prophylactic use of antifungal agents referred to in question 11, 55.6% and 52.4% of clinicians adopted the approach in patients with recent perforated intra-abdominal surgery therapy and in those with high *Candida* scores and a high rate of ICU invasive candidiasis, respectively (Table 2).

### 2.4. Participants’ Responses to Pharmacokinetic Properties of Antifungal Therapy

Appendix A (Appendix A) describes the assessment of knowledge and practice of HCPs regarding the pharmacokinetics of antifungal therapy. This section included eight questions. Pertaining to question 12, fortunately, it was found that almost four quarters (78%) of the clinicians were sure that fluconazole is the antifungal agent needed to be prescribed for a urosepsis patient with a positive *Candida albicans* urine culture. Similarly, for another scenario in question 14, where an ICU patient is at higher risk of IFI with a *Candida* score above three and a creatinine clearance (CrCl) of 30 mL/min, clinicians correctly believe that the dose of fluconazole needs to be renally adjusted (66.7%), as shown in Figure 4. Unfortunately, for question 15, a case of IC in a pregnant female, the majority of clinicians (63.5%) were uncertain which antifungal agent provides the safest option, and only 19% of them were likely to give amphotericin B (Figure 4). Furthermore, two-thirds (60%) of HCPs correctly knew that amphotericin B deoxycholate is known for its inherent renal toxicity, and 65.5% believe that echinocandins do not require renal dose adjustments, as presented in questions 16 and 17, respectively (Figure 4). In question 18, regarding the only azole that needs dose adjustment in hepatic impairment, only 20.6% of the clinicians were able to correctly identify voriconazole (Figure 4); meanwhile, 36.5% were not sure about it. Furthermore, clinicians were not confident when prescribing the right fluconazole dose for the treatment of candidemia. Only 31.7% of them were sure that fluconazole 800 mg loading dose (LD) followed by 400 mg per day was the proper dose (Figure 4). We also learned that 70% of clinicians do not rely on the oral route when administering fluconazole at their current ICU setting irrespective of the bioavailability, as shown in question 13 (Appendix A). Although fluconazole has high bioavailability and is not affected by food, HCPs’ responses can be explained by the septic status of the majority of ICU patients which dramatically affects gastrointestinal absorption.

### 2.5. Participants’ Overall Knowledge and Practices Scores

The assessment of systemic antifungal prescribing knowledge and practice among physicians and pharmacists was given in Table 2. It can be observed that the prevalence of physicians who demonstrated correct answers by starting antifungal treatment immediately when managing ICU patients with confirmed candidemia was statistically significantly higher than pharmacists (*p* = 0.003) and using a prophylactic antifungal agent in patients who had undergone a recent perforated intra-abdominal surgery (*p* = 0.016) as reported in question 9 and 11, respectively. Vice versa, pharmacists showed better correct ratings regarding antifungal agents for a pregnant woman with IC (*p* = 0.001), and the azole that needs dose adjustment in hepatic impairment (*p* = 0.023), as seen in questions 15 and 18, respectively. Furthermore, the overall pharmacokinetics knowledge score was higher in pharmacists (*p* = 0.050) (Figure 5). Considering the overall level of knowledge and practice, it was poor among 50.8% of HCPs, moderate in 46%, and only 3.2% had a good knowledge level. Interestingly, the overall knowledge scores of the ICU physicians and the clinical pharmacists (*p* = 0.925) were not significantly different and their antifungal therapy score was similar as well (*p* = 0.906) (Figure 5).

### 2.6. Participants’ Responses to Institution-Related Factors Affecting Antifungal Prescribing Practice

In Table 3, 50.8% of the HCPs indicated that their institution had antifungal sensitivity testing and 39.7% had surrogate antifungal tests such as the β-D-glucan or galactomannan test. The most prevalent *Candida* species, as reported by the institution, was *Candida albicans* (69.8%). Approximately 61.9% would de-escalate fluconazole based on the isolate sensitivity reported in culture. Nearly 60% of the HCPs reported that the average number of days for fungal culture results was 3 to 5 days. Only 27% of the HCPs use antifungal agents for the purpose of prophylaxis. The most preferred route for prescribing fluconazole in an ICU setting was the IV to oral route (60.3%).

### 2.7. Socio-Demographic Factors Affecting Participants’ Knowledge and Practices Score

In Table 4, consultants were more associated with having better knowledge and practice scores (F = 3.131; *p* = 0.032). Likewise, the private sector and other hospitals were more associated with having better knowledge and practice scores (F = 2.883; *p* = 0.032). However, the difference in knowledge and practice according to years of experience did not reach statistical significance (*p* = 0.084). In Table 5, post-hoc analysis indicates that there was a significant difference in the knowledge and practice score between consultants and residents (*p* = 0.019); however, the comparison of knowledge and practice among other job descriptions did not reach statistical significance (*p* > 0.05).

## 3. Discussion

This study was conducted to determine the knowledge and practices of clinicians regarding antifungal therapy prescribing in an ICU setting. To our knowledge, this is the first study in Saudi Arabia that investigated clinicians’ prescribing knowledge and practice toward antifungal therapy. Based on our findings, we confirmed that even regular prescribers in our area are in need of continuing education because, from this study, we observed a major gap in the knowledge and practices of clinicians in this domain.

Although most of our participants (75%) admitted to adding an empiric antifungal for patients who remain feverish despite broad-spectrum antibiotic therapy, only about one quarter (24%) were found to have started it initially in dialysis-dependent patients exhibiting signs of septic shock. About 33% of participants thought the positivity of a β-D-glucan test per se warrants the addition of antifungal therapy, and 22% thought that urinary colonization with *Candida* species is an indication to start antifungal agents even in a vitally stable patient. These practices contradict the published practice guidelines where empiric antifungal therapy is recommended in patients who have signs of septic shock and who are at risk of IC such as dialysis in our scenario [6]. It is worth mentioning that empiric therapy based solely on colonization with *Candida* species appears inadequate and false positivity remains a significant limitation of the β-D-glucan test, especially in critical illness [6].

On the other hand, clinicians demonstrated good knowledge and practices regarding initial antifungal treatment. Overall, 68.3% of them were confident that echinocandin (e.g., micafungin, caspofungin, anidulafungin) was the primary drug for the initiation of empiric antifungal treatment. Furthermore, 44.4% of them knew that the treatment should be continued for at least 2 weeks even if the patient is clinically stable. Likewise, three-quarters of the clinicians (75%) were aware that the antifungal treatment should stop if there was no clinical response to the therapy in 4–5 days. Conversely, in one study conducted by Aldrees et al [15] in an academic tertiary care center in Riyadh, Saudi Arabia to assess the appropriate utilization of caspofungin, more than half (56.3%) of the empiric treatment group continued beyond five days from initiation, even though there was no evidence of invasive *Candida* infections. The discrepancy can be explained by the fact that responding to patient scenarios in surveys differs from tracing real-world practices.

Regarding de-escalation practices, 57.1% of clinicians were sure that antifungal therapy should be altered from echinocandin to fluconazole if the patient was clinically stable and the blood culture grew isolates like fluconazole-sensitive *Candida albicans*; whereas, in the case of *Candida glabrata*, 54% of clinicians were aware that the prescription of echinocandin has to be continued until the patient improves. These findings are in accordance with clinical practice guidelines for the management of candidiasis [6]. In a retrospective study carried out in a 30-bed mixed ICU, located in the University Hospital of Lille, France, de-escalation was done in only 20% of the patients receiving empiric antifungal therapy for suspected *Candida* infection [25]. Though a higher percentage (57.1%) of our participants selected the correct approach of de-escalating echinocandin to fluconazole in *Candida albicans* susceptible strains, this percentage might have been exaggerated by the survey nature of our study; hence, it may be even lower in real-life practices. Furthermore, clinicians could not distinguish the right prescription for clinically stable patients with blood culture isolates such as voriconazole-sensitive *Candida Krusei*. Nearly 60% of them would continue with the echinocandin as long as the patient improved when, in fact, it needs to be de-escalated to voriconazole (3.4%) [6].

Comparing our results to the report of Valerio et al. [20], which was conducted in four European countries, we found some similarities; about 30% of physicians failed to distinguish *Candida* urinary infection from colonization, as did 22% in our study population. A low proportion of European physicians (41%) were able to identify the need for empiric antifungal added to antibiotics in patients with sepsis and femoral catheter-related infections. In a similar scenario of sepsis in dialysis-dependent patients in this study, we reported that only 24% of clinicians started antifungal therapy. Another survey-based study conducted in a 1550-bed tertiary care hospital in Spain identified serious gaps in the knowledge of prescribing physicians about important aspects of the diagnosis, prophylaxis, and treatment of IFIs [26].

Pertaining to invasive aspergillosis (IA), 38% of clinicians knew that voriconazole is a better antifungal agent than either liposomal amphoteric B (25.9%) or echinocandin (12.1%) for the management of IA. These results were concordant with Valerio et al.’s study [21], where 57% of the European physicians chose voriconazole as a first-line agent, as adopted by the international guidelines [27].

It is important to note that only about one-half of clinicians (51%) indicated that antifungal sensitivity tests were available in their current institution but 35% indicated that they were not available and 14% were not sure about the availability of susceptibility testing in their institutes. This was noted in a study conducted to assess the epidemiology and burden of IFI in the Arab League countries, where data concerning in vitro antifungal susceptibility testing were lacking in many of the documented available studies [28].

It can be further observed that *Candida albicans* is the most dominant *Candida* species reported by HCPs in our study (70%) and others were not dominantly reported including *Candida tropicalis* (8%) and *Candida parapsilosis* (4.8%). Literature from Saudi Arabia reported that *Candida albicans* is the most commonly detected *Candida* species [28,29,30]. However, in the study by Aldardeer et al. [4], *Candida glabrata* was the most commonly specified blood culture of *Candida* species followed by *C. albicans,* which is not consistent with our results.

The overall level of knowledge and practice was poor among 50.8% of our study HCPs, moderate in 46% and good in only 3.2%. This was concordant with a cross-sectional survey evaluating Nigerian resident doctors’ knowledge and awareness about IFIs, where only two (0.002%) out of the 1046 respondents had a good level of awareness of IFIs [22]. Moreover, the study reported statistically significant differences in knowledge about IFIs among the various cadres of doctors as the level of knowledge increased with seniority, which was also confirmed in our research as consultants had better knowledge and practice scores [22].

The utilization of systemic antifungal agents has increased significantly in most tertiary centers. However, AFS has received very little attention [26]. Assessing knowledge and tracing practices of prescribing physicians and clinical pharmacists represents the first step essential for the development of an AFS program [26]. The application of an AFS program is associated with appropriate antifungal drug use, improved resource utilization and cost savings [18,19].

### Limitations

Our study is somehow limited by the relatively small sample size of participating clinicians; however, this is partly explained by the selection of only ICU specialists in the five tertiary hospitals from which we gained approval in our area. Furthermore, access to ICU staff was greatly limited by the restrictions of the COVID-19 outbreak during which our study was conducted and even some ICU staff declined participation owing to workload. Our study was also conducted in one governorate in Saudi Arabia (Al-Ahsaa), which renders the generalization of results to other areas or other countries impractical.

## 4. Materials and Methods

### 4.1. Study Design and Setting

A cross-sectional, multi-center, survey-based study was conducted to evaluate ICU staff physicians’ and clinical pharmacists’ knowledge and practice of prescribing systemic antifungals in an ICU setting. This study was conducted among five tertiary hospitals located in Al-Ahsaa, Saudi Arabia between January and May 2021. The five hospitals were King Fahad Hospital Hofuf (KFHH), Almoosa Specialist Hospital, Al-Ahsaa Hospital, Prince Saud Bin Jalawy Hospital (PSBJ), and King Faisal Hospital.

### 4.2. Study Population and Data Collection

ICU staff physicians including consultants, specialists, and residents as well as ICU clinical pharmacists of the aforementioned hospitals were invited to complete a paper-based or electronic self-administered survey. The study investigators paid visits to the study site to meet and interview the clinicians in person. The objectives of the study were explained to the participating clinicians. Participation was voluntary and verbal consent was sought from all participants. Potential participants were selected by purposive sampling. The participating HCPs were allowed 15–20 min to complete the survey tool. Owing to the constraints imposed by the COVID-19 outbreak, especially in critical care areas, an electronic self-administered questionnaire was further provided to the ICU staff that the investigator did not have access to.

### 4.3. Questionnaire

The survey was designed by the authors in accordance with the IDSA 2016 updated Clinical Practice Guideline for the Management of Candidiasis [6]. Four of the main authors are academicians and three of them are hospital clinical preceptors in the area of critical care as well. Our survey was revised and validated by two infectious diseases and critical care professors as well as a pharmacology professor at King Faisal University. For the purpose of testing the validity, length, clarity, and comprehensibility of the survey, we piloted the questions among thirteen ICU staff physicians and two clinical pharmacists who provided their comments and amendments were subsequently made to the items of the survey. The internal consistency of the questionnaire was measured using Cronbach’s alpha (a = 0.73), which was indicative of acceptable reliability.

The survey tool had 26 items; 19 of which assessed antifungal therapy knowledge and 7 of which explored institution-related practices of the participating clinicians. The survey was divided into four parts; part 1 covered some demographic characteristics of the participants. Part 2 of the survey assessed knowledge and practice of prescribing systemic antifungals in the ICU setting with regards to the decision to initiate and agent choice for empiric, prophylactic or definitive treatment in a given patient scenario. Part 3 of the tool assessed dosing and some pharmacokinetic parameters of the most commonly prescribed antifungal agents. The last part explored some institutional factors affecting antifungal therapy prescribing.

### 4.4. Sample Size and Ethical Approval

Our study was conducted in 5 tertiary centres in our area; the total number of ICU staff in these centres was 75. We employed the Raosoft calculator; the sample size was computed at a 5% margin of error, and a 95% CI. The calculated sample size was 63, and we were able to recruit this number. Ethical IRB approvals were obtained from King Fahad Hospital (KFHH) and Almoosa Specialist Hospital in Hofuf, Al-Ahsaa. The King Fahad Hospital IRB approval extended our privilege to collect data from other MOH hospitals in Al-Ahsaa.

### 4.5. Statistical Analysis

The prescribing knowledge and practice related to systemic antifungals was assessed using 19-item questionnaires, where the correct answer for each question had been identified and had been coded with 1 and the incorrect answer had been coded with 0. Knowledge questions 1 and 10 had multiple responses with 2 and 3 correct answers, respectively, giving a total knowledge questionnaire of 22 items. The total knowledge score was obtained by adding all 22 items and a possible score range from 0 to 22 had been generated. The actual score range based on participants’ ratings has a range from 4 to 19 points.

Categorical variables were measured as frequency and proportion (%) and continuous variables were expressed as mean and standard deviation. The comparison of knowledge and practices according to the socio-demographic characteristics of HCPs was performed using an independent sample *t*-test or one-way ANOVA test. Post-hoc analysis has been carried out using the Tukey HSD test. A *p*-value of 0.05 was considered statistically significant. Normality testing was performed using the Shapiro–Wilk test as well as the Kolmogorov–Smirnov test. The data follows the normal distribution. Thus, the parametric tests were applied. All data analyses were performed using the Statistical Packages for Software Sciences (SPSS) version 26 (IBM Corporation, Armonk, New York, USA).

## 5. Conclusions

This study has revealed a significant gap in the knowledge and practice of clinicians toward prescribing antifungal therapy in our area. The overall level of knowledge and practice was poor among 50.8% of HCPs, and only 3.2% of HCPs had a good knowledge level in the present area of study. The overall knowledge and practice scores were almost identical across physicians and clinical pharmacists (*p* > 0.05), but the score of pharmacists for the pharmacokinetics domain was significantly higher compared to that of physicians (*p* = 0.05). Our findings, therefore, reflect the situation of antifungal prescription by clinicians in the region of study. These findings have exposed the need for a tailored training program essential for carrying out an antifungal stewardship program. Although our findings cannot be generalized, the knowledge gaps observed may partly explain the inappropriate use of antifungal agents, which consequently contributes to a global increase in antifungal resistance, adverse outcomes and increased costs. Compared to numerous antibiotic prescribing evaluation studies, there is a paucity of antifungal prescribing investigations making this research the first of its kind in the region. Hence, to fully evaluate this situation and utilize the applicability of our findings, further regional studies utilizing a larger population are needed in order to gain a better insight into the knowledge and practices of clinicians as regards prescribing antifungal therapy in our region.

## Figures and Tables

**Figure 1 antibiotics-12-00238-f001:**
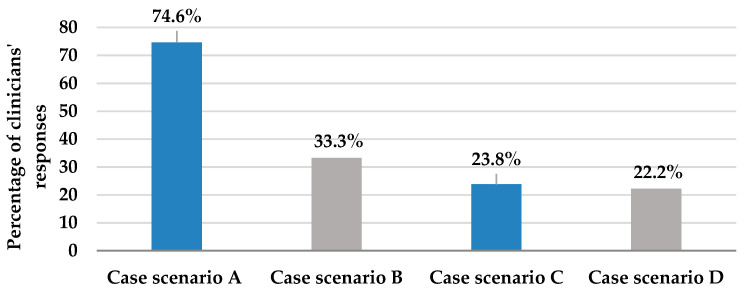
This figure describes clinicians’ responses to question 1; selection of the appropriate case scenario where immediate empiric antifungal should be started, from 4 provided patient case scenarios. Case scenario A: Patient is still febrile and did not respond to broad-spectrum antibiotics (appropriate). Case scenario B: Patient shows a positive B-D-glucan test (inappropriate). Case scenario C: Patient exhibits signs of septic shock and is on maintenance hemodialysis (appropriate). Case scenario D: Patient is vitally stable, but urine culture reveals *Candida* species growth (inappropriate).

**Figure 2 antibiotics-12-00238-f002:**
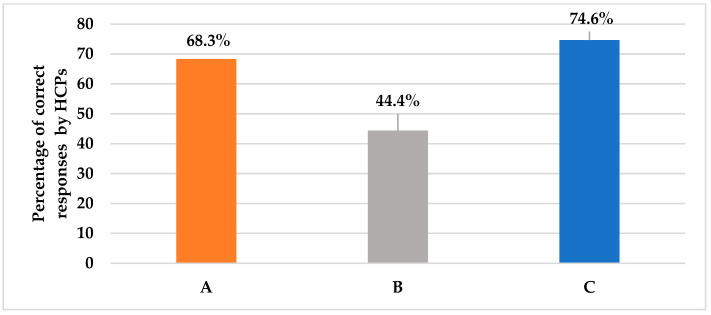
This figure shows the percentage of the HCPs’ correct answers for different case scenarios assessing empiric antifungal regimen considerations, as presented in questions 2, 4, and 5, respectively. (**A**): Choice of initial empiric antifungal in suspected invasive candidiasis; 68.3% answered echinocandins. (**B**): Duration of empiric antifungal therapy for improved and negatively cultured patients; 44.4% answered 2 weeks. (**C**): Action in case of a lack of response at 4–5 days and negative cultures; 74.6% answered stop antifungal.

**Figure 3 antibiotics-12-00238-f003:**
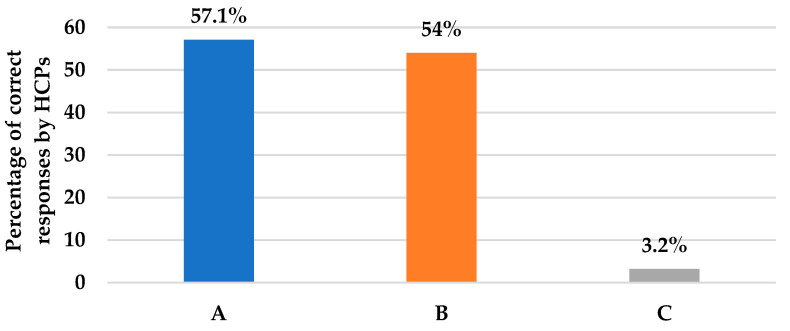
This figure illustrates the percentages of HCPs’ correct answers for questions assessing de-escalation to a targeted antifungal in three different case scenarios, as presented in questions 6, 7, and 8, respectively. (**A**): Action if culture results were C. *Albicans*; 57.1% answered de-escalation to fluconazole. (**B**): Action if culture results were C. *glabrata*; 54% answered proceeding with echinocandin. (**C**): Action if culture results were C. *Krusei*; 3.2% answered de-escalation to voriconazole.

**Figure 4 antibiotics-12-00238-f004:**
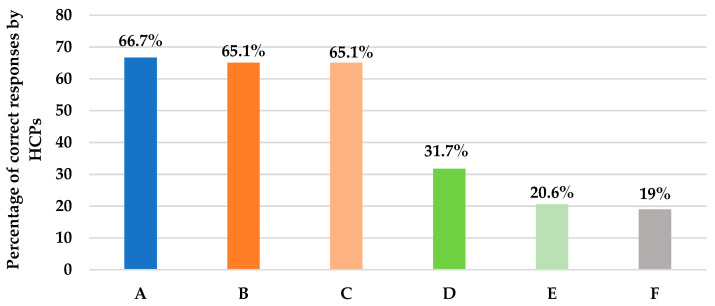
This figure describes the percentage of HCPs’ correct answers to questions assessing the pharmacokinetic properties of antifungal agents. (**A**): Action with fluconazole in CrCl < 30 mL/min; 66.7% answered to adjust the dose. (**B**): Antifungal agent with inherent renal toxicity; 65.1% answered amphotericin deoxycholate. (**C**): Class that does not require renal dose adjustments; 65.1% answered echinocandin. (**D**): Dose of fluconazole in candidemia; 31.7% answered 800 mg LD, then 400 mg. (**E**): Azole that needs adjustment in hepatic impairment; 20.6% answered voriconazole. (**F**): Antifungal agent for invasive candidiasis safe in pregnancy; 19% answered amphotericin B.

**Figure 5 antibiotics-12-00238-f005:**
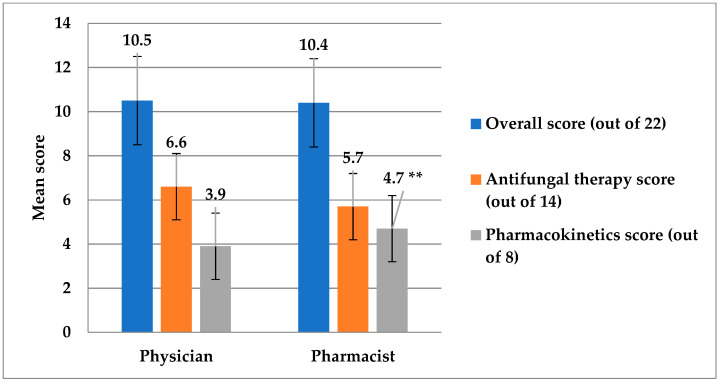
This figure shows the comparison of scores between physicians and pharmacists for the antifungal therapy domain, the pharmacokinetics domain, and the overall knowledge and practice scores. ** The score of pharmacists for the pharmacokinetics domain was significantly higher compared to physicians (*p* = 0.05).

**Table 1 antibiotics-12-00238-t001:** Socio-demographic characteristics of participants according to physician and pharmacist.

Study Variables:	Overalln (%) ^(n = 63)^	Physician n(%) ^(n = 42)^	Pharmacist n(%) ^(n = 21)^
**Workplace**			
• MOH hospital	18 (28.6%)	13 (31.0%)	05 (23.8%)
• MNGHA hospital	07 (11.1%)	01 (02.4%)	06 (28.6%)
• Private sector hospital	34 (54.0%)	26 (61.9%)	08 (38.1%)
• Other hospitals	04 (06.3%)	02 (04.8%)	02 (09.5%)
**Position:**			
• ICU consultant	06 (09.5%)	06 (14.3%)	0
• ICU assistant consultant	02 (03.2%)	02 (04.8%)	0
• ICU specialist	25 (39.7%)	25 (59.5%)	0
• ICU Resident	09 (14.3%)	09 (21.4%)	0
• ICU Clinical pharmacist	14 (22.2%)	0	14 (66.7%)
• Inpatient Hospital pharmacist	07 (11.1%)	0	07 (33.3%)
**Years of experience:**			
• <3 years	13 (20.6%)	05 (11.9%)	08 (38.1%)
• 3–5 years	09 (14.3%)	08 (19.0%)	01 (04.8%)
• 5–10 years	16 (25.4%)	10 (23.8%)	06 (28.6%)
• 10–15 years	16 (25.4%)	13 (31.0%)	03 (14.3%)
• >15 years	09 (14.3%)	06 (14.3%)	03 (14.3%)

MOH; Ministry of Health, MNGHA; Ministry of National Guard Health Affairs, ICU; Intensive Care Unit.

**Table 2 antibiotics-12-00238-t002:** Assessment of systemic antifungal prescribing knowledge and practice among physicians and pharmacists ^(n = 63)^.

Knowledge and Practice Statement Section A: Antifungal Therapy	Overall	Physician	Pharmacist	*p*-Value ^§^
	Correct Answer n(%)	Correct Answer n(%)	Correct Answer n(%)	
1. Time to immediately start an empiric antifungal therapy in an ICU patient ^†^				
• If the patient exhibits signs of septic shock and is on maintenance hemodialysis	15 (23.8%)	09 (21.4%)	06 (28.6%)	0.545
• If the patient is still febrile and did not respond to broad-spectrum antibiotics	47 (74.6%)	29 (69.0%)	18 (85.7%)	0.222
2. Before starting an empiric anti-fungal therapy, do you usually look back at the patient’s recent azole exposure?	4 (6.3%)	02 (04.8%)	02 (09.5%)	0.595
3. Whenever you suspect invasive candidiasis in a critically ill non-neutropenic patient, what empiric antifungal therapy do you usually start according to your practice?	43 (68.3%)	30 (71.4%)	13 (61.9%)	0.567
4. If the patient improved after the empiric antifungal therapy, and had stable vitals, for how long you will pursue the antifungal agent?	28 (44.4%)	17 (40.5%)	11 (52.4%)	0.427
5. If the patient had no clinical response to the empiric antifungal therapy at 4–5 days and negative follow-up cultures for fungal growth, what action is to be taken?	47 (74.6%)	34 (81.0%)	13 (61.9%)	0.130
6. Assuming you started an empiric therapy using an echinocandin like caspofungin, and the patient started improving, was clinically stable and the isolate from the blood culture was *Candida albicans*, what would your action be?	36 (57.1%)	25 (59.5%)	11 (52.4%)	0.602
7. Assuming you started an empiric therapy using an echinocandin like caspofungin, and the patient started improving, was clinically stable and the isolate from the blood culture was *Candida glabrata*, what would your action be?	34 (54%)	25 (59.5%)	09 (42.9%)	0.285
8. Assuming you started an empiric therapy using an echinocandin like caspofungin, and the patient started improving, was clinically stable and the isolate from the blood culture was *Candida krusei*, what would your action be?	2 (3.2%)	01 (02.4%)	01 (04.8%)	1.000
9. In an ICU patient with confirmed candidemia, how do you usually react?	36 (57.1%)	30 (71.4%)	06 (28.6%)	**0.003 ****
10. If you ever come across a case of invasive aspergillosis, which antifungal will you order according to your practice and availability at your institute?	24 (38.1%)	16 (38.1%)	08 (38.1%)	1.000
11. For which of the following scenarios would you use a prophylactic antifungal agent in your ICU? ^†^				
• In patients with a high *Candida* score, and a high rate of invasive candidiasis in the ICU	33 (52.4%)	21 (50.0%)	12 (57.1%)	0.789
• In patients who have undergone a recent perforated intra-abdominal surgery	35 (55.6%)	28 (66.7%)	07 (33.3%)	**0.016 ****
• In patients with necrotizing pancreatitis	11 (17.5%)	09 (21.4%)	02 (09.5%)	0.310
Antifungal therapy knowledge score (mean ± SD) ^‡^	6.26 ± 1.96	6.57 ± 1.64	5.67 ± 2.42	**0.084**
**Section B: Pharmacokinetics of Antifungal therapy #**				
Pharmacokinetics of Antifungal therapy score (mean ± SD) ^‡^	4.16 ± 1.59	3.88 ± 1.42	4.71 ± 1.82	**0.05 ****
**Overall score based on correct ratings**				
Total knowledge and practice score (mean ± SD) ^‡^	10.4 ± 2.81	10.5 ± 2.29	10.4 ± 3.69	0.925
Level of knowledge				
• Poor	32(50.8%)	21 (50.0%)	11 (52.4%)	0.906
• Moderate	29 (46%)	20 (47.6%)	09 (42.9%)	
• Good	2 (3.2%)	01 (02.4%)	01 (04.8%)	

^†^ Variable with multiple response answers. ^§^ *p*-value has been calculated using the Fischer exact test. ^‡^ *p*-value has been calculated using independent sample *t*-test. ** Significant at *p* ≤ 0.05 level. # Refer to Appendix A in Appendix A for Q12–Q19.

**Table 3 antibiotics-12-00238-t003:** Assessment of institution-related factors affecting antifungal prescribing practice.

Behavior Statement	n(%)
1. Does your institute have antifungal sensitivity testing?	
• Yes	32 (50.8%)
• No	22 (34.9%)
• I am not sure	09 (14.3%)
2. Does your institute have surrogate antifungal tests such as the β-D-glucan or galactomannan test?	
• Yes	25 (39.7%)
• No	17 (27.0%)
• I am not sure	21 (33.3%)
3. What is the most prevalent *Candida* species you come across in your practice as reported by your institute?	
• *Candida-albicans*	44 (69.8%)
• *Candida-nonalbicans*	06 (09.5%)
• *Candida-glabrata*	01 (01.6%)
• *Candida-parapsilosis*	03 (04.8%)
• *Candida-tropicalis*	05 (07.9%)
• I cannot tell, my institute does not usually report the *Candida* species	04 (06.3%)
4. If one of your ICU patients is suffering from invasive candidiasis and is receiving echinocandin as a broad-spectrum antifungal, you decided to de-escalate to fluconazole, on what basis do you carry out the de-escalation?	
• Based on the isolate sensitivity reported in the culture	39 (61.9%)
• Based on previous knowledge from the literature	07 (11.1%)
• Based on a fixed hospital protocol	05 (07.9%)
• I usually do not de-escalate if I start with an echinocandin	12 (19.0%)
5. What is the turnaround time for fungal culture results in your institute?	
• 3–5 days	36 (57.1%)
• 5–7 days	20 (31.7%)
• 7–10 days	05 (07.9%)
• >10 days	02 (03.2%)
6. Do you use antifungal agents for the purpose of prophylaxis in your institute (no active infection, but the risk of developing one is high)?	
• Yes	17 (27.0%)
• No	46 (73.0%)
7. What is the preference in your institute/practice when prescribing fluconazole in an ICU setting?	
• IV is preferred to the oral route	38 (60.3%)
• Oral route is preferred for IV	02 (03.2%)
• Both are prescribed in my institute	21 (33.3%)
• I am not sure	02 (03.2%)

**Table 4 antibiotics-12-00238-t004:** Differences in the overall score of knowledge and practice according to workplace and years of experience.

Factor	Knowledge and Practice Score (22) Mean ± SD	F-Test	*p*-Value ^§^
**Workplace**			
• MOH hospital	9.38 ± 3.39	2.883	**0.043 ****
• MNGHA hospital	9.14 ± 2.12		
• Private sector hospital	10.9 ± 2.29		
• Other hospitals	12.7 ± 3.20		
**Job description**			
• Consultant	12.2 ± 3.61	3.131	**0.032 ****
• Specialist	10.6 ± 1.25		
• Resident	8.33 ± 1.58		
• Pharmacist	10.4 ± 3.69		
**Years of experience**			
• <3 years	9.00 ± 3.51	2.169	0.084
• 3–5 years	9.67 ± 1.73		
• 5–10 years	10.8 ± 2.07		
• 10–15 years	10.6 ± 2.09		
• >15 years	12.2 ± 3.93		

MOH; Ministry of Health, MNGHA; Ministry of National Guard Health Affairs. ^§^ *p*-value has been calculated using one-way Anova test. ** Significant at *p* ≤ 0.05 level.

**Table 5 antibiotics-12-00238-t005:** Post-hoc analysis of the multiple mean differences in the score of knowledge and practice according to the job description ^(N = 63)^.

(I) Job Description	(J) Job Description	Mean Difference (I–J)	Sig.
Consultant	Specialist	1.610	0.455
	Residents	3.917 *	0.019
	Pharmacist	1.869	0.342
Specialist	Consultant	−1.610	0.455
	Residents	2.307	0.130
	Pharmacist	0.259	0.988
Residents	Consultant	−3.917 *	0.019
	Specialist	−2.307	0.130
	Pharmacist	−2.048	0.230
Pharmacist	Consultant	−1.869	0.342
	Specialist	−0.259	0.988
	Residents	2.048	0.230

Post-hoc analysis has been conducted using Tukey’s HSD test. * Significant at *p* ≤ 0.05 level.

## Data Availability

The data that support the findings of this study are available on request from the corresponding author [S.I.].

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
