# Peer review of "Evaluation of Systemic Antifungal Prescribing Knowledge and Practice in the Critical Care Setting among ICU Physicians and Clinical Pharmacists: A Cross-Sectional Study"

_antibiotics, 2023, doi:10.3390/antibiotics12020238_

Round 1

Reviewer 1 Report

The main issue of the manuscript is its structure. The needed parts of a research paper are mixed in a unnatural matter. I cannot imagine how it could happened to ended like this.

Introduction

·       The theoretical background is too weak, which is reflected in only 27 references. Please provide more insides.

·       After identifying the gap, the research aim should be outlined.

Results

·       Table 1: number of respondents is noted N or n?

·       The section include only data in tables and figures, with minimum analysis.

·       Table 5: there is no need to include all data provided by SPSS output, but it should systematised and presented just what is important for the analysis.

4. Materials and Methods

·       Please describe the study population, sampling procedure and data collection process. Also comment on the representativeness of the sample of 63 respondents.

·       The data collection instrument was adopted from previous studies, or your own. Please comment on this issue.

Conclusions

·       In conclusions section should be betted outlined the research implications, limitations and directions for future research.

Reviewer 2 Report

The relevance of this study is of a high degree, because there is a problem of increasing the incidence of fungal infections.

On the whole, this manuscript is well done and written. However, it should be noted that the way the results are presented overwhelms the reviewer. The manuscript contains very large tables with data and figures that are not entirely clear for perception. Figure captions are misleading; there is, for example, figure 2: there are inscriptions on the horizontal axis (Choice of initial empiric antifungal, Duration of empiric antifungal therapy, Action in case of lack of response to antifungal therapy), on the figure itself there are inscriptions with a different meaning (Echinocandin, 2 weeks, stop antifungal). Perhaps the authors should have made the description of the figures more detailed and understandable.

Author Response

Please see the attachment,

Reviewer 3 Report

Dear Editor,

General Comments

Many thanks for inviting me to review this paper. This study investigated antifungal prescribing knowledge of Physicians and Clinical Pharmacists in ICU settings. I write my suggestions below.

I believe this study aligns with the scope of the journal. Antibiotics is a highly reputable academic journal and has a distinguished audience. And its’ audience deserve high-quality and exquisite publications.

To increase the impact of the paper and readability authors should work how to present novelty and impact of the paper. There are some major limitations of the study. I believe this study should be reassessed after the major problems resolved.

·                I believe the title is suitable and adheres with the content of the study.

·                The novelty of the study is questionable. Since the sample size is quite low generalizability is difficult even for local areas (Such as Saudi Arabia).

·                I would like to suggest author to reduce information about AFS programs and give more details in methods and results section.

·                In abstract I would like to see the results of the reliability analysis of the questionnaire.

·                Keywords: I would like to recommend the adhere MeSH headings.

·                I believe the introduction section is well organized and beneficial.

·                Please adhere the journal guideline especially for reference sections. Need to adhere to the guidance for authors.

·                It would be better to give up to date references.

·                Did you use any checklist or guideline for the design of the study such COREQ, CROSS etc. It would be useful to see a flowchart and one of the checklists as a supplementary table.

·                How the sample size calculated. What is the a priori power score of this study?

·                Even though sample size is focused on a specific HCW the sample size is rather small. In abstract section and the main text there are a lot of generalizability of the result I believe which is not possible under these conditions.

·                More information needed about the settings and participants in methods section.

·                Shapiro-Wilk test usually fails with sample size over 30 I would like to suggest authors use Kolmogorov-Smirnov test in addition to Shapiro-Wilk test for exploration of normality.

·                There is huge need for the questionnaire development. The given information is insufficient. As a reader I would like to gather more information about the questionnaire. Was that developed by the authors, validated, or translated and cross culturally adapted?

·                The authors showed that there is a knowledge gap about AF treatments but what are the driving factors for this gap? Lack of competencies, educational deficits, motivational deficits etcetera. I believe the driving factors should be explored as well.

·                The take home message is quite broad. I believe detailed outcomes should be underlined.

·                I would like to suggest authors to give some details about the outcomes. The outcome should be explored in detail as well as findings.

·                Discussion is quite long and hard to follow. Please sum it up accordingly.

·                Limitation should be more explored.

Author Response

Please see the attachment,

Reviewer 4 Report

The current manuscript regards a cross sectional study assessing the knowledge of physicians and pharmacists on systemic antifungal prescribing. It is overall interesting, produces predictable results, and the methodology is reasonably sound. Before publication, I would like to see the following alterations:

- The introduction section should be extended, to include information on similar studies that have been done in other countries, and the respective conclusions;

- Figure 1 should be improved: what “percentage” is the Y axis referring to? The x axis has too much text, you should divide by categories, and then insert in the caption what those categories correspond to; also the caption also needs more descriptive information; same goes for the other figures;

- The “Materials and Methods” section should come before the “Results” section;

- Comment more on the limitations of the current study, not only in what concerns the small number of participants (which you already have), but also and the fact that the study was only conducted in Saudi Arabia population, hence not being possible to draw generalized conclusions for other countries.

Round 2

Reviewer 1 Report

Each table should be referred within text.

The research implications should be better outlined.

Reviewer 3 Report

Dear Editor,

General Comments

Many thanks for inviting me to review this paper. This study investigated antifungal prescribing knowledge of Physicians and Clinical Pharmacists in ICU settings. I write my suggestions below.

I believe this study aligns with the scope of the journal. Antibiotics is a highly reputable academic journal and has a distinguished audience. And its’ audience deserve high-quality and exquisite publications.

To increase the impact of the paper and readability authors should work how to present novelty and how to increase impact of the paper.

After revision the paper looks much better. However, the novelty and scientific soundness is still the major issue. I believe this study is just adequate enough to be published. 
